# Ladder-Residual: Parallelism-Aware Architecture for Accelerating Large Model Inference with Communication Overlapping

**Muru Zhang** [* 1 2]   **Mayank Mishra** [* 3]   **Zhongzhu Zhou** [1 4]   **William Brandon** [5]   **Jue Wang** [1]   **Yoon Kim** [5]
**Jonathan Ragan-Kelley** [5]   **Shuaiwen Leon Song** [1 4]   **Ben Athiwaratkun** [1]   **Tri Dao** [1 6]

## Abstract

Large language model inference is both memory-intensive and time-consuming, often requiring distributed algorithms to efficiently scale. Various model parallelism strategies are used to partition computation across multiple devices, reducing memory load and computation time. However, using model parallelism requires communication of information between GPUs, which limits the gains obtained by scaling up the number of devices. We introduce Ladder Residual, a simple architectural modification applicable to all residual-based models that enables straightforward overlapping to hide the latency of communication. **Our insight is that in addition to system optimizations, the model architecture can also be redesigned to decouple communication from computation**. While Ladder Residual can allow communication-computation decoupling in conventional parallelism patterns, we focus on Tensor Parallelism in this paper, which is particularly bottlenecked by its heavy communication. For a Transformer model with 70B parameters, applying Ladder Residual to all its layers can achieve 29% end-to-end wall clock speedup at inference time with sharding over 8 devices. We train a 1.2B and 3.5B Ladder Residual based Transformer models from scratch and observe comparable performance to a standard dense transformer baseline. We also show that it is possible to convert parts of the Llama-3.1 8B model to our Ladder Residual architecture with minimal accuracy degradation by only retraining for 3B tokens.

## 1. Introduction

With the rapid scaling of Large Language Models (LLMs) (Smith et al., 2022; Workshop et al., 2023; Brown, 2020), the compute and memory requirements for training and inference have grown significantly. Tensor parallelism (TP) (Shoeybi et al., 2020) is a widely used model parallelism technique that partitions the weights and intermediate activations across multiple GPUs. In contrast to pipeline parallelism (Narayanan et al., 2021) and data parallelism (Li et al., 2020), which rely on processing independent batches of user requests on each device, TP enables multiple devices to cooperate to process a single request, therefore in theory allowing infinite scaling given a sufficient number of processors. However, TP requires synchronizing the partitioned intermediate activations across the GPUs. This synchronization is a blocking `AllReduce` operation on the activations across the GPUs and is therefore bottlenecked by the network communication latency. Even for GPUs connected via fast interconnects (like NVLink (NVIDIA Corporation, 2024)), the communication costs can account for 38% of the latency at inference time when running a 70B transformer with batch size 4 and TP world size of 8.

Past works have attempted to overlap the communication latency of TP by overlapping computation and communication. Chang et al. (2024) write fused kernels for `AllGather` followed by matmul and matmul followed by `ReduceScatter`. They break down matmuls into tiles hide the latency of communicating a tile with the computation of subsequent tiles. Jangda et al. (2022) propose CoCoNet, a domain-specific language to express distributed machine learning workloads. They propose to generate efficient GPU kernels for computation and communication using a custom compiler for the DSL. This approach has limited applicability with existing frameworks like PyTorch (Paszke et al., 2019) and JAX (Frostig et al., 2018) since the user needs to be well-acquainted with the DSL to generate efficient GPU kernels. Moreover, with the breakneck pace of accelerator and interconnect changes, these low-level systems optimizations require a rewrite for every new generation of hardware. However, there is a fundamental limit to how much communication latency can be reduced

---

[1]Together AI [2]University of Southern California [3]MIT-IBM Watson Lab [4]University of Sydney [5]Massachusetts Institute of Technology [6]Princeton University. Correspondence to: Muru Zhang <muruzhan@usc.edu>, Mayank Mishra <mayank31398@gmail.com>, Tri Dao <tri@tridao.me>.

*Proceedings of the 42nd International Conference on Machine Learning*, Vancouver, Canada. PMLR 267, 2025. Copyright 2025 by the author(s).

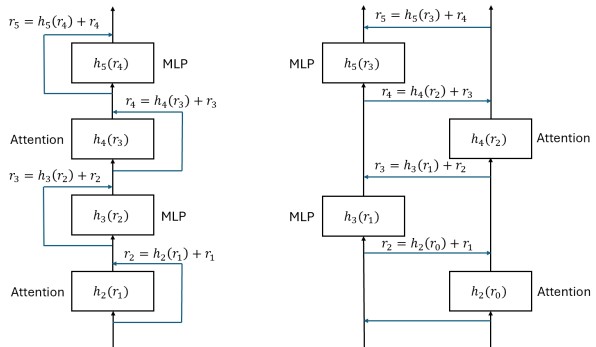

*Figure 1.* Illustration of a standard Transformer block (left) and a Ladder Residual block (right). The blue edge denotes the residual connection. In Ladder Residual, the residual connection remains the same while each module $h_i$ takes the stale input $r_{i-2}$.

with these approaches which do not change the underlying model architecture. Instead of pure hardware optimizations (e.g., larger NVLink domain connecting 36 or 72 NVIDIA Blackwell GPUs) or pure low-level software optimization (e.g., rewriting all matmuls to overlap with communication), we explore model architectural changes that would enable a reduction in communication latency while maintaining accuracy. This makes our approach quite simple to apply in practice using a high level machine learning framework like PyTorch (Paszke et al., 2019) or JAX (Frostig et al., 2018) without writing any low-level device code.

In current LLMs, communication is blocking because there is a sequential structure between communication and computation in existing model designs: we wait for communication in order to prepare the correct input for the next computation. In the prevalent residual-based architectures, the computation flow can be written as $x_{i+1} = h_{i+1}(x_i) + x_i$, where $x_i$ is the residual stream after layer $i$ and $h_{i+1}$ is the computation at layer $i + 1$. Notice that the communication of $x_i$ needs to be done before executing $h_{i+1}$ if $h_i$ is partitioned across devices. Liu et al. (2023b) found that activation changes slowly in Transformer, as the norm of each update $h_{i+1}(x_i)$ is small compared to the residual. Based on this observation, we hypothesize that maintaining the regular residual connection is enough to restrict the representation shift, and we can feed each module a "stale" input to create overlapping opportunities.

We propose *Ladder Residual*, a simple change where we reroute the residual stream after module $i - 1$ (instead of module $i$) as input to module $i+1$: $x_{i+1} = h_{i+1}(x_{i-1})+x_i$. With this design, the computation of $h_{i+1}$ is decoupled from the communication of $x_i$, enabling straightforward overlapping to hide the latency of communication. Figure 1 shows how Ladder Residual can be applied to the Transformer architecture. At inference time with TP world size of 8,

*Table 1.* Inference speedup from applying Ladder Residual on a Transformer model. The test setup is 1024 prompt length, 512 generated tokens, batch size 4, and TP world size of 8 for 1B, 3B, 8B, 34B, 70B, 176B model and TP world size of 16 for 405B model (using two nodes). The models are designed according to Llama model families (and Bloom-176B). The speedup value is calculated by comparing Ladder Transformer with Standard Transformer's inference throughput in tokens per second. We measure the speedup both with and without P2P communication (modern server nodes usually support P2P communication). Note that disabling P2P communication significantly increases the `AllReduce` latency.

| Model size | P2P disabled | P2P enabled |
|---|---|---|
| 1B | 1.39x | 1.56x |
| 3B | 1.50x | 1.57x |
| 8B | 1.40x | 1.46x |
| 34B | 1.47x | 1.44x |
| 70B | 1.59x | 1.29x |
| 176B | 1.54x | 1.35x |
| 405B | 1.57x | 1.31x |

running a 70B Transformer with Ladder Residual can be around 30% faster than the standard Transformer. In Table 1, we provide the inference speedup on Transformers of different sizes. The proposed Ladder Residual method can also be used to accelerate other forms of parallelism, although we focus on Tensor Parallelism in this paper as it is particularly bottlenecked by the heavy communication. Our method obtains 5-7% training speedup when training an 8B model with 8k context length on 64 H100s with 3D parallelism across the GPUs (Tensor Parallel, Sequence Parallel and Fully Sharded Data Parallel (FSDP) (Zhao et al., 2023; Rajbhandari et al., 2020)), but we decide to focus on the inference speed ups since training with pure FSDP is usually faster because weights can be pre-fetched and gradient synchronization using `ReduceScatter` can be overlapped in FSDP making communications in FSDP non-blocking.

Because of the widespread use of Transformer (Vaswani, 2017) based Language models, we focus on applying Ladder Residual on Transformer models in this paper, and we call the resulting model as Ladder Transformer. However, it should be noted that Ladder Residual is compatible with any residual based model architecture. We conduct experiments under two scenarios to verify if we can maintain the same performance as standard Transformer:

- **Pretraining from scratch**: We train a 1.2B and a 3.5B parameter Ladder Transformer model with 100B tokens on the FineWeb-edu dataset (Lozhkov et al., 2024) and compared it with the standard transformer of the same size trained on the same amount of tokens. We find that the Ladder Transformer matches the performance of the standard Transformer model.

- **Post-training adaptation**: We take the pretrained

Llama-3.1-8B-instruct model (Dubey et al., 2024) and apply Ladder Residual on the upper half layers. We then fine-tune it on 3B tokens to adapt to the representation shift. With this relatively light retraining, we can obtain a hybrid Ladder Llama that is on par with the original Llama model across a variety of multiple-choice and generative tasks.

In our paper, **we propose to change the model architecture to accelerate model parallelism without touching low-level kernels, making it easily deployable on any hardware**. We show that such architecture modification performs on-par with the standard transformer. As model size grows, multi-gpu or even cross-node serving will become more and more important, and our research provides a fresh perspective on designing a model architecture with parallelism optimizations in mind. Such design can be applied to any architecture that suffers from blocking communication, although in this paper we conduct experiments on Transformer-based language models due to their widespread use and popularity.

## 2. Tensor Parallelism Background

Tensor parallelism (TP) (Shoeybi et al., 2020) is a widely used technique in distributed training/inference. It partitions weights and activations across devices and performs partial computations on each device. Consider a sequence of 2 linear layers with weight matrices $A$ and $B$ and input activation $X$ that is running on 2 GPUs (TP world size of 2), we split $A$ along the output dimension into $[A_1, A_2]$, and split $B$ along the input dimension into $\begin{bmatrix} B_1 \\ B_2 \end{bmatrix}$. Then the output of the sequence of the 2 linear layers can be computed as $(XA)B = (XA_1)B_1 + (XA_2)B_2$ and we effectively partition the computation on the two devices. The final summation requires an `AllReduce` operation to aggregate the partial sums on each device, which introduces communication overhead. The `AllReduce` overhead increases with increasing message size and increasing number of devices participating in the `AllReduce`. A transformer layer consists of an attention block and an MLP block: both can be considered as a sequence of two matrix multiplications and therefore fit into the tensor parallelism paradigm described above. Thus each transformer layer contains 2 `AllReduce` operations: one for attention and another for MLP. Denoting the input to the $i^{th}$ block as $x_{i-1}$, the transformer can be viewed as the following sequential structure:

$$
\begin{aligned}
x_i^* &= h_i(x_{i-1}) \\
x_i &= \texttt{AllReduce}(x_i^*) + x_{i-1} \\
x_{i+1}^* &= h_{i+1}(x_i) \\
x_{i+1} &= \texttt{AllReduce}(x_{i+1}^*) + x_i
\end{aligned}
\tag{1}
$$

where the $*$ denotes a partial-sum that requires an `AllReduce` to replicate full output across all the GPUs.

A Transformer with N layers needs to perform the `AllReduce` 2N times and this can account for 38% of the inference latency for a 70B model using TP world size of 8, even with NVLink interconnect. If P2P communication is disabled, `AllReduce` latency can account for over 50% of the end-to-end latency. Modern nodes connect GPUs via NVLink but usually have no more than 8 GPUs per node, due to limited PCIe lanes, power density and cooling constraints in datacenters. There is a steep falloff in communication bandwidth and sharp increase in latency when the communication happens across nodes either over InfiniBand or Ethernet thus making scaling TP practically infeasible outside a node.

---

**Algorithm 1** Ladder Transformer Layer with Tensor Parallelism. Note that the `AsyncAllReduce` (**ARR**) returns a handle which is passed to the next layer.

---

1: **function** LAYER(residual, attn_out, mlp_out,
            attn_work, mlp_work)
2:    attn_work.wait()
3:    residual ← residual + attn_out
4:
5:    attn_out ← AttentionNorm(residual)
6:    attn_out ← Attention(attn_out)
7:    attn_out, attn_work ← **AAR**(attn_out)
8:
9:    mlp_work.wait()
10:   residual ← residual + mlp_out
11:
12:   mlp_out ← MLPNorm(residual)
13:   mlp_out ← MLP(mlp_out)
14:   mlp_out, mlp_work ← **AAR**(mlp_out)
15:
16:   **return**   residual, attn_out, mlp_out,
17:   attn_work, mlp_work
18: **end function**

---

## 3. Ladder Transformer

In this section, we introduce the Ladder Residual architecture when applied to Transformer and benchmark its efficiency under various model sizes and generation setups.

### 3.1. Architecture description

In Equation 1, the `AllReduce` operation is blocking the next block from execution since $h_{i+1}$ requires $x_i$ as the input. Ladder Residual mitigates this problem by routing the $x_{i-1}$ to block $h_{i+1}$, effectively making the input of $h_{i+1}$ independent of the output of the `AllReduce`, therefore allowing overlapping `AllReduce`($x_i^*$) with $h_{i+1}$.

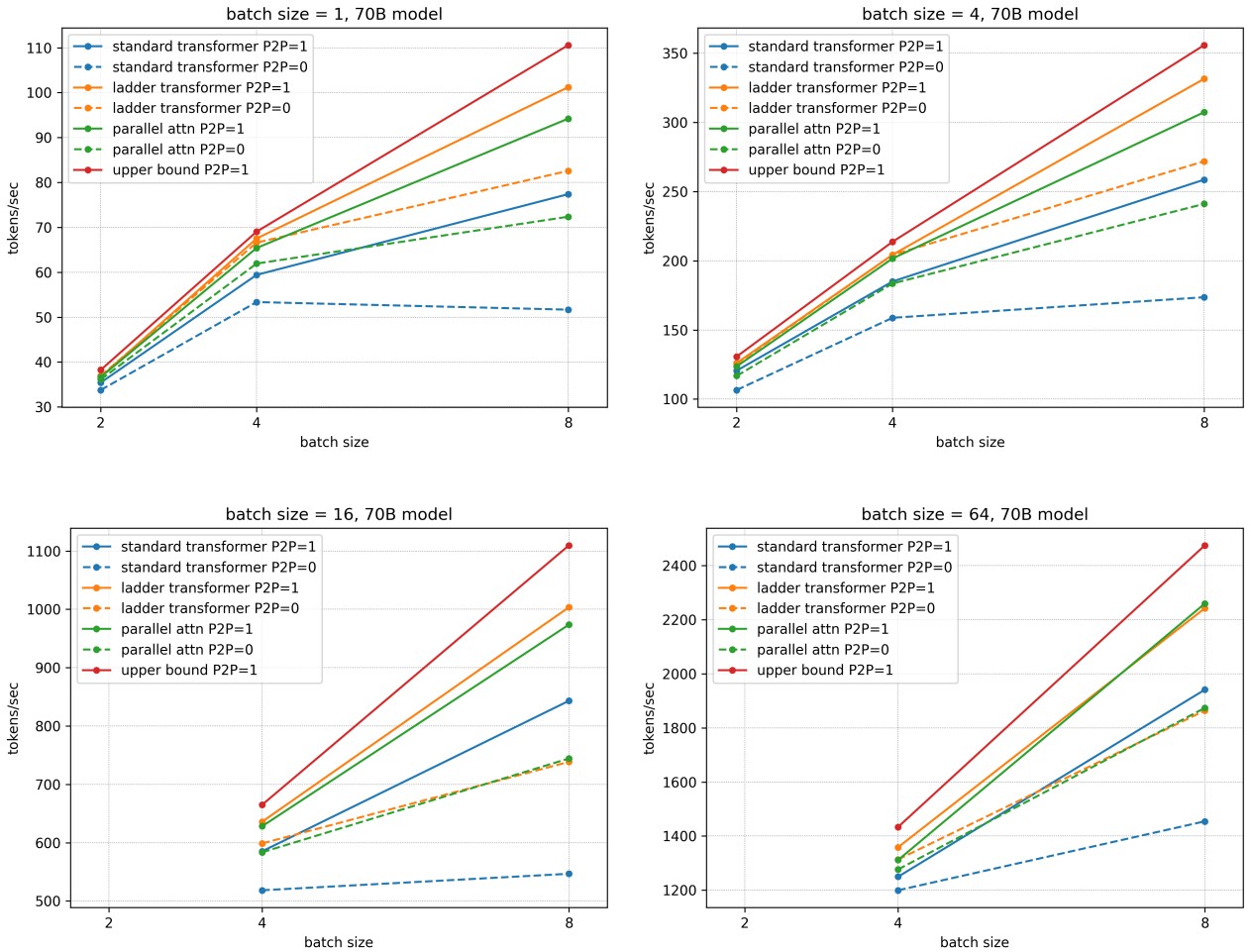

*Figure 2.* Improvement in end-to-end inference throughput achieved by communication-efficient architectures relative to a standard transformer, benchmarked on Llama-3 70B. *Standard* refers to the regular Llama-3, and *Ladder* is Llama-3 with our Ladder Residual architecture. Ladder Residual architecture can achieve up to $29\%$ greater throughput than the standard Transformer. With slower communication (P2P disabled or P2P=0), we observe speedups up to $60\%$. All experiments were conducted on a generation task with 1024 prompt tokens and 512 completion tokens. Missing data points indicate CUDA OOM.

Specifically, we change the computation flow of Equation 1 into:

$$
\begin{aligned}
x_i^* &= h_i(x_{i-2}) \\
x_i &= \texttt{AllReduce}(x_i^*) + x_{i-1} \\
x_{i+1}^* &= h_{i+1}(x_{i-1}) \\
x_{i+1} &= \texttt{AllReduce}(x_{i+1}^*) + x_i
\end{aligned}
\tag{2}
$$

Note that the residual stream of each block still takes the output from the previous block as usual, this ensures block $i$ can still process information from all previous $i-2$ blocks.

### 3.2. Inference Implementation

**Ladder Residual Implementation**: We present the Ladder Transformer's layer's pseudo-code in Algorithm 1.

To implement the Ladder Transformer, following the description of Equation 2 we call `AsyncAllReduce` for the `Attention`'s output. This returns a handle that can be used to synchronize the output to ensure that the `AsyncAllReduce` has finished. It should be noted that NCCL collectives in PyTorch always run on a different CUDA stream than the default compute stream used by PyTorch thus making them asynchronous. As soon as the `AsyncAllReduce` for Attention is called, we synchronize the previous layer's MLP's output by calling wait on the previous layer's MLP's `AllReduce` handle and subsequently the CPU launches the kernels for `MLPNorm` and then `MLP` on the default compute stream and eventually calling the `AsyncAllReduce` for MLP. The handles for these NCCL operations are then passed onto the next layer

*Table 2.* Detailed breakdown for prefill latency, decode latency and generated token/sec improvement (%) for 70B model. The speedup (%) is calculated by using latency of optimized model divided by original model. All the experiments are based on batch size 1, TP world size of 8 GPUs.

| Model | Prefill Latency Improvement (%) | Decode Latency Improvement (%) | Token/sec Improvement (%) |
|---|---|---|---|
| UpperBound-Llama-70B (P2P=1) | 30.54 | 30.00 | 42.90 |
| Parallel-Llama-70B (P2P=1) | 5.42 | 18.04 | 21.75 |
| Ladder-Llama-70B (P2P=1) | 5.78 | 23.71 | 30.79 |
| UpperBound-Llama-70B (P2P=0) | 35.84 | 52.71 | 110.7 |
| Parallel-Llama-70B (P2P=0) | 14.92 | 28.73 | 40.07 |
| Ladder-Llama-70B (P2P=0) | 6.94 | 37.71 | 59.87 |

which uses them for synchronization when needed.

**Alignment with Real-World Scenarios**: To evaluate the practical benefits of Ladder Residual, we integrated this mechanism into a standard Llama-like Transformer. Building upon `gpt-fast` (PyTorch Labs, 2024), we partition the weights of the attention and feedforward modules for tensor parallelism to optimize inference speed. We use CUDA graphs (Coleman, 2020) via PyTorch compile (with "reduce-overhead" mode) to generate static computation graphs for both the prefill and decode phase to reduce CPU kernel launch overheads which can be a big bottleneck especially during the decode phase of Transformer inference. PyTorch compile also additionally helps in accelerating inference and reducing the memory footprint for inference by emitting more efficient kernels.

### 3.3. Faster Inference with Ladder Residual

In this section, we benchmark Ladder Residual under various scenarios and show that across various model sizes, batch sizes, and TP world sizes, Ladder Transformer can obtain considerable speedup over the standard Transformer. We also benchmark under the case where P2P communication is disabled (for testing in a scenario with increased `AllReduce` latency), and show that our method can obtain more than 50% speedup. Finally, we consider cross-node Tensor Parallelism, which can be necessary for serving large models like Llama 3.1 405B, and show that Ladder Residual can improve inference throughput by more than 30%.

#### 3.3.1. SETUP

We benchmark several algorithmic variants to evaluate their performance in large-scale language model inference. The candidates include:

- Standard Transformer: The standard transformer implementation.

- Parallel Attention and MLP: Following the PaLM parallelization strategy (Chowdhery et al., 2022; Wang & Komatsuzaki, 2021), we fuse the weights of the query, key, value, gate, and up projections into a single matrix. The outputs are then split, and the attention and swiglu

are performed in parallel. While not proposed for accelerating Tensor Parallelism, we realize this architecture can effectively cut half of the communication, with the extra benefit of being able to fuse attention and mlp together, therefore we consider it as an alternative to Ladder Residual.

- Ladder Residual: The architectural optimization we propose to overlap computation with communication required for Tensor Parallelism.

- Communication-Free Upper Bound: An upper bound that removes all communication operations in the model to represent the theoretical maximum speedup achievable.

Note that different Transformer-based models usually have slightly different designs. We choose to use Llama-3's model architecture since it's one of the most widely used models. The benchmarking results here are a good representative of a variety of different design choices since the communication patterns are mostly the same across various transformer variants.

To simulate various inference scenarios, we select multiple experimental configurations. The prompt length and generation length is fixed to 1024 and 512 respectively, while we vary the tensor parallel world sizes among 1, 2, 4 and 8, and batch sizes among 1, 4, 16 and 64 to understand performance under different generation settings. All benchmarks are done on NVIDIA H100 GPUs.

To evaluate the impact of hardware communication capabilities, we attempt to simulate the performance in the presence of slow interconnects. We set `NCCL_P2P_DISABLE=1` to disable the point-to-point communication. This significantly slows down the NCCL communications and allows us to assess the performance of difference algorithms in varying communication environments. We observe a significant slowdown (for 8 GPUs) for `AllReduce` when disabling P2P communication. In our experiments and figures, we refer to this setting as P2P=0 or P2P disabled. Unless otherwise stated, P2P is always enabled (or P2P=1).

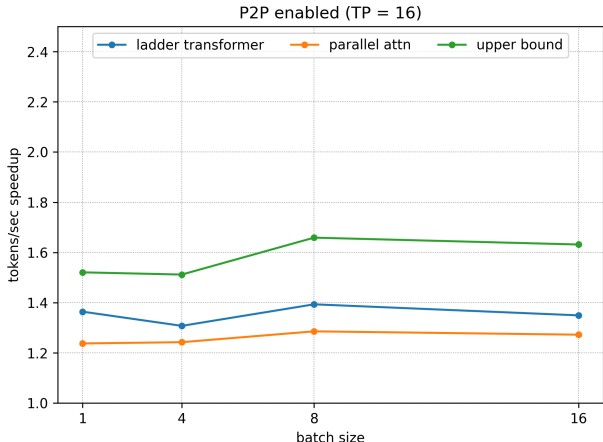

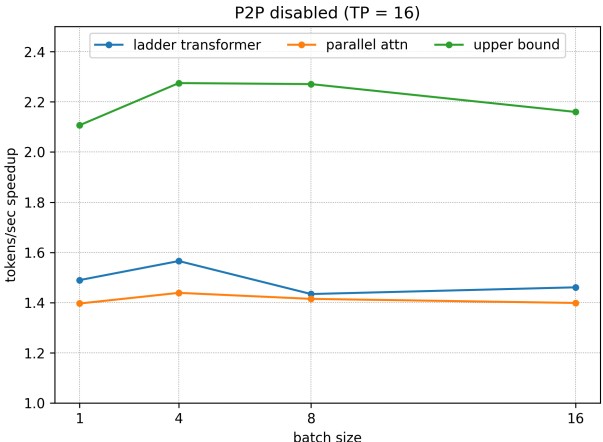

*Figure 3.* End-to-end inference throughput improvement on Llama-3-405B on a generation task with 1024 prompt tokens and 512 completion tokens. Here we use TP size 16 across two nodes each with 8 GPUs, connected with InfiniBand. Due to the high cost of cross-node communication, Ladder Residual architecture is able to achieve more than 30% improvement across various batch sizes with P2P enabled and around 50% with P2P communication disabled.

### 3.3.2. BENCHMARKING

We characterize the inference efficiency improvements enabled by Ladder Residual in three different ways.

First, we measure the **best latency** achievable (batch size 1, TP degree 8) using both the Ladder Residual architecture and a standard transformer baseline. We report end-to-end latency broken down by inference phase (*prefill* vs *decode*) in Table 2. In this latency-optimized regime, both with and without NVLink, Ladder Residual outperforms the parallel attn-mlp alternative in both prefill and decode latency.

Second, we measure the **throughput** across different TP world size and batch sizes for 70B model in Figure 2. In these throughput-oriented experiments, we again find that

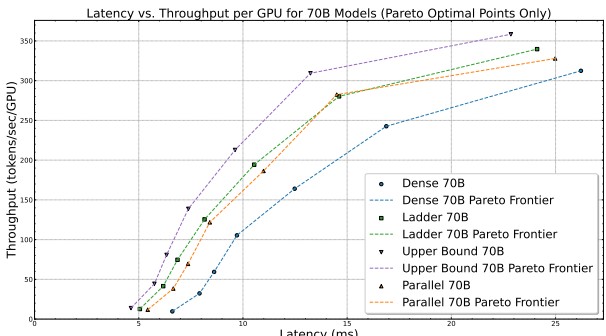

*Figure 4.* Pareto frontier of completion latency vs aggregate throughput per GPU for different 70B-scale model architectures in a batched inference setting. For each architecture, we sweep over both batch size and TP world size to find the Pareto-optimal configurations. Using less TP size results in higher throughput while using a higher TP size optimizes the latency, both have its use-case and we found that ladder architecture achieves Pareto improvements over the standard transformer architecture and the parallel transformer. All experiments measure end-to-end time on a generation task with 1024 prompt tokens and 512 completion tokens per sequence.

the Ladder Residual architecture significantly outperforms the standard transformer architecture, and that these improvements are larger when communication is slower. The throughput gains from adopting the Ladder Residual architecture increase as the TP degree increases, reflecting the greater proportion of run time spent in communication relative to compute as we partition the computation across a larger number of devices. Lastly, as shown in Table 1, the amount of improvement decrease as we increase from 8B to 70B when running with P2P communication, as the computation scales faster than communication. However, the trend is reversed when running without P2P communication, due to the much higher cost of communication in that scenario.

Finally, we consider serving model that is too large to be loaded on one node. Cross-node TP communication is very expensive, however the common practice of using intra-node TP with cross-node Pipeline Parallelism (PP) is dependent on batch size to reduce gpu idle time (for example, with batch size = 1, half of the GPUs will be idle at any given time). With the speedup from Ladder Residual, cross-node TP can be a viable option. We benchmarked 405B under such setting in Figure 3, found that even for nodes with fast InfiniBand interconnect, Ladder Residual can achieve more than 30% throughput improvement across batch sizes.

## 4. Experiments and Results

We empirically verify our assumption that applying Ladder Residual does not hurt the performance. We show that Ladder Residual can be either used when training from

scratch, or be applied to a pre-trained model with hybrid adaptation, and in both cases, the performance is on par with the original architecture.

## 4.1. Training From Scratch

We train a 1.2B and 3.5B Ladder Transformer from scratch and compare its performance with an equally sized standard Transformer model. All our models are trained on 100B tokens of FineWeb-edu dataset (HuggingFaceFW, 2024) using the StarCoder tokenizer (Li et al., 2023b). We also compare our model with the Parallel Attention/MLP architecture (Chowdhery et al., 2022; Wang & Komatsuzaki, 2021) which parallelizes the computation of the attention and the MLP module. This effectively reduces the communication cost by 50% for the tensor parallel `AllReduce` in both the forward and backward computation.

### 4.1.1. EXPERIMENTAL DETAILS

We use DDP (Distributed Data Parallel) (Li et al., 2020) to train the 1.2B and HSDP (Hybrid Sharded Data Parallel) (Zhao et al., 2023; Rajbhandari et al., 2020) to train the 3.5B models. For HSDP, we shard the model within 1 node (equipped with 8x H100 GPUs) and replicate the model outside the node. We use mixed precision training (Micikevicius et al., 2018) in BF16 (Kalamkar et al., 2019) with gradient accumulation and gradient `AllReduce`/`ReduceScatter` in FP32 for training stability. We train all our models with 2048 context length with a batch size of 4M tokens in a batch. The models are trained with cosine scheduler with a warmup of 8B tokens to a peak learning rate of $3 \times 10^{-4}$. The learning rate is then decayed over 92B tokens to $3 \times 10^{-5}$.

We use EleutherAI's LM eval harness (Gao et al., 2024) to evaluate models on ARC (Clark et al., 2018), HellaSwag (Zellers et al., 2019), PIQA (Bisk et al., 2020), SciQ (Welbl et al., 2017) and Winogrande (Trinh & Le, 2018). We also evaluate perplexity on Wikitext (Merity et al., 2017).

### 4.1.2. RESULTS

The full results can be found at Table 3. We find that at the 1.2B model scale, Ladder Transformer achieves performance similar to Standard Transformer while beating the Parallel Transformer. At 3.5B parameter scale, we find that Standard Transformer is better than Ladder Transformer model with 3.2% lower perplexity and 1.2 points of absolute difference in accuracy. The Parallel Transformer has almost the same performance as Ladder Transformer at the 3.5B scale.

## 4.2. Post-training adaptation

We investigate the feasibility of directly applying Ladder Residual on an existing pre-trained model. We applied Ladder Residual to the upper half of the Llama-3.1 8B Instruct to keep the performance since we found that touching the lower layers can destroy knowledge that is hard to recover without large-scale retraining. We evaluate the adapted models on 8 benchmarks across a range of domains: accuracy on MMLU (5-shots) (Hendrycks et al., 2021) and ARC-Challenge (ARC-C, 25-shots) (Clark et al., 2018), normalized accuracy on OpenBookQA (OBQA) (Mihaylov et al., 2018), HellaSwag (HS, 10-shots) (Zellers et al., 2019), and TruthfulQA (TQ, mc1) (Lin et al., 2022). exact-match accuracy on GSM8K(GSM, 8-shots) (Cobbe et al., 2021), pass@1 on HumanEval+(HE+) (Chen et al., 2021), aggregated accuracy on IFEval (Zhou et al., 2023), and length controlled win rate (Dubois et al., 2024) against gpt4-turbo on AlpacaEval (Li et al., 2023c). The evaluation of HumanEval+ is conducted with EvalPlus (Liu et al., 2023a), AlpacaEval is done with the AlpacaEval2 library, and the rest of the evaluations are conducted with the LM-Evaluation-Harness library (Gao et al., 2024).

### 4.2.1. EXPERIMENTAL DETAILS

We convert a state-of-the-art open-source model, Llama-3.1-8B-Instruct into a hybrid Ladder Residual structure, by applying Ladder Residual to the upper half of the model (layers 16-32 for LLaMA-3.1-8B-Instruct). We call this variant Hybrid-Ladder-8B-16L in Table 4. We also experiment with more aggressive adaptation where we applied Ladder Residual to the layers 12-32 and we call this experiment Hybrid-Ladder-8B-20L. We conduct supervised fine-tuning (SFT) for the resulting model on the 7M subset and the Gen subset of the Infinity-Instruct dataset[1], which contains 3B tokens. We train for 2 epochs with AdamW optimizer with a batch size of 32. We use $5 \times 10^{-6}$ learning rate with 200 steps of linear warmup, followed by cosine annealing to the end. We use Axolotl[2] for our SFT experiments and Open LM Engine[3] for our pretraining experiments.

As shown in Table 4, after adaptation, there is a huge performance drop mainly on generative tasks as the computation flow is messed up. But after light retraining, the hybrid Ladder Llama is able to reach the same level of performance with the original Llama. By applying Ladder Residual on the last 16 layers, we can obtain 21% end-to-end wall clock speed up for inference with TP world size of 8 and batch

---

[1] https://huggingface.co/datasets/BAAI/Infinity-Instruct
[2] https://github.com/axolotl-ai-cloud/axolotl
[3] https://github.com/open-lm-engine/lm-engine

*Table 3.* Performance of three architectures under two sizes, trained on FineWeb-edu for 100B tokens.

| Model | ARC-C | ARC-E | HellaSwag | PIQA | SciQ | Winogrande | Average | Wikitext PPL |
|---|---|---|---|---|---|---|---|---|
| Standard-Transformer-1.2B | 34.22 | 70.33 | 41.10 | 71.49 | 87.30 | 55.41 | 59.98 | 18.54 |
| Parallel-Transformer-1.2B | 30.46 | 67.97 | 40.35 | 71.16 | 87.40 | 55.17 | 58.75 | 18.95 |
| Ladder-Transformer-1.2B | 31.31 | 67.76 | 41.18 | 71.49 | 86.60 | 55.17 | 58.92 | 18.42 |
| Standard-Transformer-3.5B | 38.99 | 74.12 | 46.48 | 74.59 | 92.00 | 58.48 | 64.11 | 14.48 |
| Parallel-Transformer-3.5B | 38.48 | 73.02 | 45.55 | 73.67 | 90.00 | 57.46 | 63.03 | 14.96 |
| Ladder-Transformer-3.5B | 36.77 | 72.43 | 45.66 | 73.72 | 89.90 | 58.96 | 62.91 | 14.90 |

*Table 4.* All models are either LLama-3.1 models or are adapted from Llama-3.1 8B Instruct in this table. Performance comparison across various benchmarks. Zeroshot denotes directly applying Ladder Residual without any retraining. $n$L denotes that $n$ layers of the Llama-3.1-8B-Instruct are adapted with Ladder Residual.

| Model | MMLU | ARC-C | OBQA | HS | TQ | GSM | HE+ | IE | AE | Average |
|---|---|---|---|---|---|---|---|---|---|---|
| Llama-3.1-8B-Instruct | 68.14 | 60.32 | 43.00 | 80.04 | 36.84 | 84.99 | 60.40 | 52.57 | 18.69 | 56.11 |
| Hybrid-Ladder-8B-16L-zeroshot | 63.19 | 56.57 | 42.60 | 77.70 | 35.50 | 10.54 | 30.50 | 46.25 | 11.99 | 41.65 |
| Hybrid-Ladder-8B-16L-retrained | 65.93 | 59.13 | 42.20 | 78.86 | 39.66 | 80.29 | 59.10 | 59.02 | 21.95 | 56.24 |
| Hybrid-Ladder-8B-20L-retrained | 62.31 | 59.90 | 42.60 | 77.49 | 36.72 | 76.19 | 48.80 | 59.05 | 21.72 | 53.86 |

size of 1. Our results demonstrate the potential of Ladder Residual being a drop-in adaptation technique to make the model faster without sacrificing performance. We additionally experiment with applying Ladder Residual to the last 20 layers of Llama and found that it leads to a slight drop in performance. There is a chance that with longer adaptation, or smarter adaptation techniques like distillation or iterative training, we can obtain a Ladder-Llama with more layers adapted. We leave the further exploration to future work.

## 5. Discussion

### 5.1. Compatibility with other Parallelism Techniques

It should be noted that Ladder Residual is fully compatible with other model parallelism techniques like Pipeline Parallelism (PP). The `AllReduce` communication for TP is still asynchronous except at the PP boundary where the `AllReduce` needs to be waited upon to complete. Once the `AllReduce` result is available, we can forward 3 tensors (`residual`, `current_mlp_output` and `current_attention_output`) to the next pipeline stage. It should be noted that this is still pretty cheap since generally the P2P communication during inference is latency bound and can be implemented easily using the `batch_isend_irecv` API in PyTorch. Our approach also seemlessly works with Distributed Data Parallel (DDP) and Fully Sharded Data Parallel (FSDP).

### 5.2. Comparison to a 30% larger Ladder Transformer

Since the 70B Ladder Tranformer model achieves 30% higher tokens/sec compared to the 70B standard transformer as shown in Table 2, we compare the standard transformer to a 30% larger ladder transformer in Table 5. We find that the 30% larger ladder transformer is better than the standard transformer at both the 1.2B and 3.5B scale on average ac-

curacy across bechmarks and wikitext perplexity while still achieving a much higher inference throughput.

## 6. Related Work

**Communication overlapping in parallelism** Overlapping communication has been a widely explored area in prior works in order to achieve higher performance for distributed training. For Tensor Parallelism, prior works (Jangda et al., 2022; Wang et al., 2022; NVIDIA, 2023) decompose the communication into more fine-grained operations in order to find computations with no dependency to overlap. Our work doesn't rely on such decompositions and therefore doesn't require Sequence Parallelism to handle the partitioned activations before all-gather. In FSDP (FairScale authors, 2021), the all-gather communication is usually prefetched to be overlapped with the communication. Pipeline Parallelism (NVIDIA, 2023; Li et al., 2023a; Lamy-Poirier, 2023) on the other hand, chunks the data into mini-batches which creates more opportunity for overlapping. Compared to these other parallelism approaches, TP has the advantage to be independent of the batch size or sequence length, and can partition the computation as much as possible given enough GPUs in theory.

**Efficiency-aware architecture improvements** Prior works have explored various alternative designs for Transformer, for example parallel attention and mlp (Chowdhery et al., 2022; Wang & Komatsuzaki, 2021), linear attention (Katharopoulos et al., 2020), Grouped Query Attention (Ainslie et al., 2023), Cross-Layer Attention (Brandon et al., 2024) to improve the training and inference efficiency. Some of these variants are more widely adopted than others, due to the degree of impact they have on performance and efficiency. Past works have also considered adapting an existing checkpoint to these efficient variants. Ainslie et al.

*Table 5.* Performance of the Standard Transformer compared to a 30% larger Ladder Transformer at 1.2B and 3.5B scale, trained on FineWeb-edu for 100B tokens.

| Model | ARC-C | ARC-E | HellaSwag | PIQA | SciQ | Winogrande | Average | Wikitext PPL | Tokens/sec |
|---|---|---|---|---|---|---|---|---|---|
| Standard-Transformer-1.2B | 34.22 | 70.33 | 41.10 | 71.49 | 87.30 | 55.41 | 59.98 | 18.54 | 1008.29 |
| Ladder-Transformer-1.5B | 33.96 | 70.16 | 42.58 | 71.98 | 87.90 | 55.41 | 60.33 | 17.47 | 1277.66 |
| Standard-Transformer-3.5B | 38.99 | 74.12 | 46.48 | 74.59 | 92.00 | 58.48 | 64.11 | 14.48 | 949.6 |
| Ladder-Transformer-4.5B | 40.96 | 75.00 | 46.81 | 73.99 | 90.80 | 57.70 | 64.21 | 14.05 | 1217.71 |

(2023) extracted grouped-query attention from a multi-head attention model, and Wang et al. (2024) converted a Llama model to a Mamba model by retraining on 50B tokens to close the performance gap. Comparatively, our adaptation is much lighter (3B tokens), showing that the representation shift introduced by Ladder Residual is easier to recover. Wang et al. (2024) considered converting a Llama model to a Mamba model and used distillation to retrain the converted model. Such training paradigms that specifically tune the model to align with the original model could further improve the Ladder Residual based models.

## 7. Conclusion

We introduce Ladder Residual, architectural modifications that allow overlapping communication with computation for model parallelism. We show that when running Tensor Parallelism, Ladder Residual can achieve great speed up across various model sizes, batch sizes, and number of GPUs. When applying Ladder Residual to Llama-3.1 8B Instruct, we only need lightweight retraining to reach the same level of performance as the original model while being 21% faster, showing its potential to be a plug-in for any pretrained Transformer. We also trained a 1.2B and 3.5B Ladder Transformer from scratch, and found that they are comparable to the standard Transformer of the same size while achieving over 55% speedup. Given that such a simple architectural change can obviate the need for expensive interconnects while maintaining model quality, we hope that our method will inspire even closer co-design between model architecture and inference systems.

## Impact Statement

This paper presents work whose goal is to advance the field of Machine Learning. There are many potential societal consequences of our work, none which we feel must be specifically highlighted here.

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

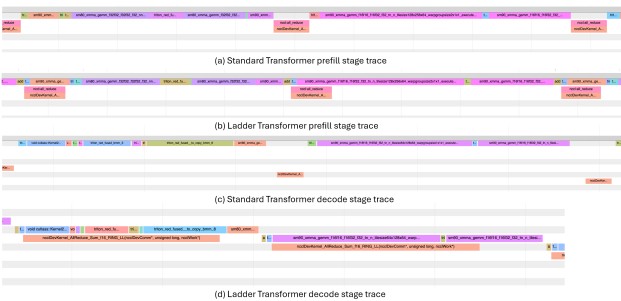

(a) Standard Transformer prefill stage trace

(b) Ladder Transformer prefill stage trace

(c) Standard Transformer decode stage trace

(d) Ladder Transformer decode stage trace

*Figure 5.* Traces generated by PyTorch Profiler. As shown in the plot for Standard transformer the NCCL operations block the computation whereas in Ladder Transformer the NCCL operations can be overlapped with the computation.

# A. Appendix

## A.1. PyTorch Profiler Trace

Here we provide the trace generated by the PyTorch Profiler[4] in Figure 5.

---

[4] https://pytorch.org/tutorials/recipes/recipes/profiler_recipe.html

