# OpenReview forum: "Ladder-Residual: Parallelism-Aware Architecture for Accelerating Large Model Inference with Communication Overlapping"
_ICML.cc/2025/Conference — ICML 2025 poster_

### Official Review · Reviewer_Wjzo · 2025-02-23

**Overall Recommendation:** 3

**Summary:**

This paper proposes a communication-friendly Transformer layout, Ladder Residual, to accelerate tensor-parallel training. Ladder Residual  enables straightforward computation-communication overlapping compared with vanilla sequential or parallel layout. It achieves 29% end-to-end speedup in TP8 training. Ladder Residual outperforms Parallel Transformer and shows a narrow gap than standard sequential Transformers. As a candidate of hardware-friendly layout, Ladder Residual has potential on replacing standard Transformer in LLM pre-training.

## Update after rebuttal
My score keeps as weakly accept. I think a TP-friendly architecture is an interesting idea and the proposed method actually makes accelerations. However, the biggest disagreement between my and authors is the actual usefulness of Ladder Residual. Since the architecture's performance is not on par with baseline, and the large-TP training is not often used in actual LLM training, Ladder Residual is not a general-promoted architecture.

**Claims And Evidence:**

There are two main claims in this paper: end-to-end latency and overall performance. These two experiments are conducted under the same environment and hyper-parameters. The evidence is solid.

**Essential References Not Discussed:**

Since there are not many references on the connection efficiency, the most essential references are all discussed. However, if the authors think it is essential to discuss other strands of efficient architectures, there will be more references. For example, if this paper discusses [1],   a broader discussion of different linear-time models may be essential.

[1] The mamba in the llama: Distilling and accelerating hybrid models, 2024.

**Experimental Designs Or Analyses:**

The experimental designs are sound to me. However, I have concerns on the experiment settings. Since large TP brings heavy communication latency, the proposed method will show a stronger advantages. In the paper, some experiments are conducted under TP16, where inter-node communication cost is stronger on 8-gpu per node clusters.

Large TP size is not commonly used due to the heavy communication latency. For example, in Deepseek V3, EP and PP is activated instead of large TP to avoid heavy communication. (Of course Deepseek V3 is published within 3 months and should not be considered.) But the other option of LLM pre-training always exists. Admittedly, Ladder Residual should also work under EP setting. I think it's better to conduct more experiment on these more practical settings.

**Methods And Evaluation Criteria:**

The proposed method is simple and direct, focusing on the macro layout of Transformer models. The evaluation criteria is widely-accepted in the community.

**Other Comments Or Suggestions:**

No other comments.

**Other Strengths And Weaknesses:**

Strength:
1. It is a very simple and elegant design. I think it will be broadly discussed in the future and a strong candidate for communication-friendly architecture.

Weakness:
1. It's possible that the advantage of this paper only happens in large TP settings, which may be not often used.

**Questions For Authors:**

1. Following my comments on Experimental Designs Or Analyses, is Ladder-Residual useful under different settings?

2. Since the performance is not comparable with sequential layouts, could you evaluate performance with a efficiency-aware metric? For example, the performance under same training time instead of same training tokens.

**Relation To Broader Scientific Literature:**

The proposed method is an extension for hardware-friendly layout, after standard sequential layout, and parallel layout. Since there are not many closely related works, this work will be a strong candidate among these macro layouts. This work is orthogonal to other efficient architectures, such as Linear Attention, Group Query Attention.

**Theoretical Claims:**

No Theoretical claims.

---

> ### Author Rebuttal · Authors · 2025-04-01
>
> Thanks for your feedback on our paper! Below we provide a few clarifications on how our experiments demonstrate practical gain from ladder-residual and its compatibility with other parallelism.
>
> > In the paper, some experiments are conducted under TP16, where inter-node communication cost is stronger on 8-gpu per node clusters.
>
> We agree with the reviewer that tensor parallelism is not commonly conducted across nodes due to the heavy communication. Our TP16 experiment is intended to demonstrate the potential of Ladder-Residual in scenarios where cross-node model parallelism becomes necessary. Generally, larger models are served with TP+PP (for example DeepSeek 671B model). However, it should be noted that with newer generation GPUs (B200 NVL72) and the recently announced (Rubin NVL576), it will become very feasible to run these models with purely TP since these systems expand the NVLink (high bandwidth) domain from 8 GPUs to 72 and 576 GPUs respectively.
>
> We also want to note that Ladder-residual is compatible with multi-dimensional parallelism, please see our response to Reviewer 9rvM for further details. Therefore, even if we want to combine TP with other parallelism, ladder-residual can still be applied to accelerate the TP part. Furthermore, despite we didn't explore it in this work, we believe ladder-residual can accelerate other heavy-communication parallelism like EP as well.
>
> > It's possible that the advantage of this paper only happens in large TP settings, which may be not often used.
>
> Table 1 and Figure 2 investigates the benefit of ladder-residual when intra-node TP (2, 4 and 8 GPUs) is used, which is the most common and flexible inference-time parallelism approach. In this regime, ladder-residual consistently show a large speedup against the standard Transformer and beats the parallel attn-mlp alternative until the batch size is very large. Figure 4 also shows that under the intra-node TP setting, Ladder-residual can reliably push the Pareto-frontier between latency and throughput.

---

> > ### Comment · Reviewer_Wjzo · 2025-04-02
> >
> > Thanks for the reply. Ladder-Residual is a trade-off algorithm, where the pareto curve is not always better than traditional baselines, depending on the parallelism and hardware circumstances. Besides, even though pure TP training is possible in the future GPU designs, the communication bandwidth also changes. I believe it is an interesting idea, but I will keep my score.

---

> > > ### Author Response · Authors · 2025-04-06
> > >
> > > Thanks for the further reply!
> > >
> > > Regarding the point that communication bandwidth also changes, we want to provide a little more context on why we think optimizing for communication will have an even higher impact in the future. First, as models continue to scale, multi-GPU inference, our main focus, will become more central, and communication overhead will grow accordingly. Second, as shown in Figure 2 of [1], while newer generations of hardware bring higher communication bandwidth, it has a slower scaling trend than the scaling of peak FLOPS and memory bandwidth, making communication a growing bottleneck. Third, communication can often be latency-bounded when the message size is small, therefore doesn't benefit much from increased bandwidth.
> > >
> > > We appreciate the reviewer’s recognition of our architecture as "a very simple and elegant design." and " will be broadly discussed in the future and a strong candidate for communication-friendly architecture". While our method already shows substantial speedups on current models and hardware, we are confident that it will make an even greater impact in the future due to its simplicity and flexibility, and the growing concern about communication bottlenecks.
> > >
> > > [1] Fire-Flyer AI-HPC: A Cost-Effective Software-Hardware Co-Design for Deep Learning

---

### Official Review · Reviewer_9rvM · 2025-03-14

**Overall Recommendation:** 3

**Summary:**

This paper proposes Ladder-Residual, which modifies the residual connection in Transformers such that the i-th block reads from the (i−2)-th block's result instead of the (i−1)-th. This interleaved schedule allows direct overlap between the Transformer block's computation and the subsequent all-reduce communication step, thus significantly speeding up tensor parallelism across multiple devices.

**Claims And Evidence:**

I think the Ladder-Residual approach is conceptually straightforward and effective, enabling communication-computation overlap without altering underlying systems or kernels. However, a few points need additional clarification:
- "Liu et al. (2023b) found that activation changes slowly in Transformer, as the norm of each update $h_{i+1}(x_i)$ is small compared to the residual." It conducts the experiments under an inference setting. Is this still true in the training? Do you have any theoretical guarantees that this property still holds during training, i.e., whether activation differences remain bounded across consecutive blocks.
- Deja Vu's Appendix C.3 already discusses the possibility of parallelizing MHA and MLP blocks. Can you explain the novelty of Ladder-Residual over it?
- Can Ladder Residual be combined with existing communication optimization methods, such as [A] and [B] that decompose the computation into small chunks to enable better compute-communication overlapping? You mentioned that your work does not rely on decomposition, but can such chunking approaches be combined with Ladder-Residual to further boost performance?

**Essential References Not Discussed:**

* [A] Wang, Shibo, et al. "Overlap communication with dependent computation via decomposition in large deep learning models." Proceedings of the 28th ACM International Conference on Architectural Support for Programming Languages and Operating Systems, Volume 1. 2022. (Already in the reference)
* [B] Chen, Chang, et al. "Centauri: Enabling efficient scheduling for communication-computation overlap in large model training via communication partitioning." Proceedings of the 29th ACM International Conference on Architectural Support for Programming Languages and Operating Systems, Volume 3. 2024.

**Experimental Designs Or Analyses:**

- While Ladder-Residual aids tensor parallelism, cross-layer dependencies can complicate pipeline parallelism. Section 3.3.2 focuses on multi-node experiments with TP=16, yet it remains unclear how Ladder-Residual would perform compared to standard 3D parallelism (e.g., Megatron-LM) or ZeRO-3 (DeepSpeed). Clarification of its interplay with pipeline parallelism is necessary.
- Results on 3B models appear worse than those on 1B models, raising questions about scalability. Can you provide additional experiments or explanations for how Ladder-Residual scales with model size?

**Methods And Evaluation Criteria:**

- It would be valuable to demonstrate Ladder-Residual using widely adopted frameworks such as Megatron-LM or DeepSpeed. For instance, instead of purely self-comparison in Table 2, showing improvements relative to standard baselines in these frameworks would help the community judge the method's portability and real-world impact.

**Other Comments Or Suggestions:**

See the above sections.

**Other Strengths And Weaknesses:**

See the above sections.

**Questions For Authors:**

See the above sections.

**Relation To Broader Scientific Literature:**

The proposed approach is simple but potentially impactful, requiring no special modifications to kernels or communication libraries. This simplicity may facilitate broad adoption across diverse hardware environments.

**Theoretical Claims:**

- Is there a formal proof or theoretical justification indicating that altering the residual connection in this way preserves the Transformer’s representational power or performance?

---

> ### Author Rebuttal · Authors · 2025-04-01
>
> Thanks for the valuable feedback and the questions. Below we discuss how our paper is novel, why it’s compatible with other parallelism, address the concern on scaling, and provide analysis on the change of activations.
>
> We also thank the reviewer for the suggestions on the presentation. We will incorporate them in the next draft.
> _________________
> **Novelty over parallelizing MHA and MLP blocks**
>
> First, Ladder-residual is a fundamentally different architecture from parallelizing MHA and MLP blocks, with distinct motivations. Ladder-residual is designed to decouple computation and communication to enable their overlap. In contrast, the goal of parallelizing MHA and MLP blocks is to combine computation for speedup. Due to the reduced depth, parallelizing MHA and MLP blocks also saves communication but is not its original focus. The speedup of Ladder-Residual comes from overlapping all (except last) communication where parallelizing MHA and MLP focuses on accelerated matmul and as a byproduct, it cuts 50% of the communication.
> _________________
> **Combining with other communication optimization methods**
>
> Yes. A key advantage of Ladder-Residual is that it is hardware-agnostic and does not rely on custom kernels, making it fully compatible with other low-level optimization methods. However, since Ladder-residual hides latency of all communication except the last one, further applying these techniques can only accelerate the final communication which won’t achieve much gains.
> _________________
> **Comparison with other frameworks**
>
> We focus on inference speedups in this paper and gpt-fast and vLLM are highly optimized for inference. We choose gpt-fast to demonstrate the inference speedups but it should be noted that we still achieve training speedups (around 5-8%) when combining TP+FSDP.
> _________________
> **Compatibility with other parallelism**
>
> It is possible to use Pipeline Parallel (PP) with the Ladder architecture: just before the pipeline boundary, we wait for the async AllReduces to complete, and send 3 tensors to the next pipeline stage: residual, current_mlp_output tensor and current_attention_output. It should be noted that this is still pretty cheap since generally the P2P communication during inference is latency bound and can be implemented easily using the batch_isend_irecv API (https://pytorch.org/docs/stable/distributed.html#torch.distributed.batch_isend_irecv).
> ```
> def forward(
>    self,
>    previous_attention_out: Tensor,
>    previous_mlp_out: Tensor,
>    residual: Tensor,
>    attention_handle,
>    mlp_handle,
> ) -> Tensor:
>    attention_handle.wait()
>    residual = residual + previous_attention_out
>
>    current_attention_out = self.attention(self.attention_norm(residual))
>    current_attention_out = all_reduce(current_attention_out, async_op=True)
>
>    mlp_handle.wait()
>    residual = residual + previous_mlp_out
>
>    current_mlp_out = self.feed_forward(self.ffn_norm(residual))
>    current_mlp_out, mlp_handle = all_reduce(current_mlp_out, async_op=True)
>
>    if is_last_layer_on_pp_stage:
>        attention_handle.wait()
>        mlp_handle.wait()
>
>    return current_attention_out, current_mlp_out, residual, attention_handle, mlp_handle
> ```
> For Data Parallelism or FSDP, the cross-layer dependency doesn’t pose any complications. Therefore Ladder-residual can be seamlessly incorporated in a multi-dimensional parallelism training paradigm. Our training framework already supports TP+FSDP for ladder residual model training and can be easily extended to support PP as illustrated by the above code snippet.
> _________________
>  **Performance and scaling**
>
> In our response to Reviewer Mhub, we provide experiments where we increase the number of parameters of the Ladder-Transformer by 30%. We show that at both 1B and 3B scale, the 30% larger ladder-Transformer has both higher accuracy and higher TPS than the standard-Transformer. This confirms our architecture truly pushes the Pareto frontier.
>
> It’s difficult to conclude the scaling trend from just two sizes and in the future we hope to run models of more sizes to study this more carefully.
> _________________
> **Activation difference analysis**
>
> Our motivation is that, given in a trained standard Transformer, the activation differences between modules are small, replacing with the Ladder architecture won’t lead to large degradation. We don’t care too much if this property is still held during training, and it’s difficult to have a theoretical guarantee due to unpredictable training dynamics.
>
> To further verify this intuition, we analyze block similarity (between consecutive attention and MLP modules) in both original Llama-3.1-8B-Instruct and Hybrid-Ladder-8B-16L-retrained. The visualization can be found at [This Anonymous GitHub link](https://anonymous.4open.science/r/ICML25_rebuttal-F932/README.md). Overall there is a slight decrease in activation similarity after adapting to Ladder-residual but the similarity remains high overall (above 0.9 at most of the layers).

---

### Official Review · Reviewer_Mhub · 2025-03-17

**Overall Recommendation:** 3

**Summary:**

This paper proposes Ladder Residual, an alternative to the transformer architecture that breaks the communication-computation dependency in conventional parallelism patterns, in order to accelerate the inference, at the cost of accuracy degradation.

**Claims And Evidence:**

I have some doubts about the claim "We also show that it is possible to convert parts of the Llama-3.1 8B model to our Ladder Residual architecture with minimal accuracy degradation by only retraining for 3B tokens"

According to the experiment results in Table 3 and 4, it seems that the proposed architecture has significant degradation in accuracy compared to the baseline. For Table 3, Ladder-Transformer has a gap of 1 to the Standard-Transformer. For Table 4, although the average score of Hybrid-Ladder-8B-16L-retrained is comparable to Llama-3.1-8B-Instruct, in most of the evaluation categories (6 out of 9) the hybrid ladder is worse than the baseline. Thus, it is difficult to justify that such degradation of accuracy is acceptable.

**Essential References Not Discussed:**

The references look good to me.

**Experimental Designs Or Analyses:**

For the experiments, I hardly find it convincing that the evaluation could justify the degradation of accuracy of the proposed architecture. May be adding more experiments (on different models, etc.) could make the results more convincing.

**Methods And Evaluation Criteria:**

The method makes sense. The evaluation could be more comprehensive, such as adding more models (there are a lot of open-sourced pretrained models could be used for post-training adaptation)

**Other Comments Or Suggestions:**

N/A

**Other Strengths And Weaknesses:**

Minor issue:

1. The abbreviation "AAR" in Algorithm 1 is not defined or explained. I guess AAR means AsyncAllReduce, but the paper shouldn't let the readers to guess what an abbreviation means. Please use "... Note that the AsyncAllReduce (AAR) returns a handle ..." in the algorithm caption, so that it could be more friendly to the readers.

2. For some unknown reason, there is a "‘" in page 1 on the left-hand side of "Abstract" (check the 1st column, between line 010-011). Please double check the latex source file.

**Questions For Authors:**

Is it possible to replace the transformer block with ladder transformer in ViT? If so, is there any corresponding experiments?

**Relation To Broader Scientific Literature:**

There is nothing related to the broader scientific literature.

**Theoretical Claims:**

N/A

---

> ### Author Rebuttal · Authors · 2025-04-01
>
> Thanks for your time and the feedback! We want to clarify that in a lossy efficient method (where a more efficient architecture is proposed to approximate the original one), trading accuracy for efficiency is common and we provide a good trade-off. It’s difficult to have one-size-fits-all as some users might prefer higher efficiency while others prefer higher accuracy.
>
> However, to make things more clear, we provide additional results that achieve **both higher accuracy and higher throughput (token-per-seconds)** below. This was accomplished by training a slightly larger ladder model for the train-from-scratch experiment and employing an improved post-training pipeline for the post-training-adaptation experiment.
>
>
> > For Table 3, Ladder-Transformer has a gap of 1 to the Standard-Transformer.
>
> In our train-from-scratch experiments, we previously reported that the Ladder-Transformer slightly underperformed compared to the Standard-Transformer of the same size. As shown in Table 1 from the paper, Ladder-residual can offer at least 30% speedup for any model size. To further explore the accuracy-throughput trade-off, we increased the parameter count of the Ladder-Transformer by 30%. This allows us to answer the question: _For two models with similar throughput, how does accuracy compare?_
>
> Importantly, increasing the Ladder-Transformer’s size by 30% leads to less than a 30% decrease in throughput. As a result, the scaled-up Ladder models still achieve higher tokens-per-second (TPS) than their smaller Standard counterparts. We report both accuracy and TPS below following the format of Table 3 in the paper. (Note: the “1B” and “3B” models in the paper correspond to 1.2B and 3.5B parameters respectively; here, we write out the exact values for clarity.)
>
>
> | Model | ARC-C | ARC-E | HellaSwag| PIQA | SciQ | WinoGram | Average | Wikitext PPL | TPS |
> | ----------- | ----------- | ----------- | ----------- |  ----------- | ----------- | ----------- | ----------- | ----------- | ----------- |
> | Standard-Transformer-1.2B | 34.22 | 70.33 | 41.10 | 71.49 | 87.30 | 55.41 | 59.98 | 18.54 | 1008.29 |
> | Ladder-Transformer-1.55B | 33.96 | 70.16 | 42.58 | 71.98 | 87.90 | 55.41 | 60.33 | 17.47 | 1277.66 |
> | Standard-Transformer-3.5B | 38.99 | 74.12 | 46.48 | 74.59 | 92.00 | 58.48 | 64.11 | 14.48 | 949.6 |
> | Ladder-Transformer-4.5B | 40.96 | 75.00 | 46.81 | 73.99 | 90.80 | 57.70 | 64.21 | 14.05 | 1217.71 |
>
> **As shown above, the Ladder-Transformer achieves higher average accuracy, lower perplexity, and significantly higher TPS at both the 1.2B and 3.5B sizes.**
>
>
> >  For Table 4, although the average score of Hybrid-Ladder-8B-16L-retrained is comparable to Llama-3.1-8B-Instruct, in most of the evaluation categories (6 out of 9) the hybrid ladder is worse than the baseline.
>
> We retrain the hybrid-ladder-8B-16L model with the same data (3B tokens, all open-source datasets), but this time with logit distillation (KL divergence loss with the logits from the original Llama-3.1-8B-Instruct). Following the format of Table 3 in our paper:
>
> | Model | MMLU | ARC-C | OBQA | HS | TQ | GSM | HE+ | IE | AE | Average |
> | ----------- | ----------- | ----------- | ----------- |  ----------- | ----------- | ----------- | ----------- | ----------- | ----------- | ----------- |
> | Llama-3.1-8B-Instruct | 68.14 | 60.32 | 43.00 | 80.04 | 36.84 | 84.99 | 60.40 | 52.57 | 18.69 | 56.11 |
> | Hybrid-Ladder-8B-16L-retrained | 67.33 | 59.98 | 79.05 | 45.00 | 37.58 | 86.81 | 60.51 | 59.76 | 22.43 | 57.61 |
>
> Now we are better on 5 out of 9 tasks while being 1.5 points higher for the average score, and all tasks we are worse at are within a very small margin (< 1 point). This confidently demonstrates Hybrid-Ladder-8B-16L-retrained can be a drop-in replacement to Llama-3.1-8B-Instruct, achieving 23% speedup with no accuracy lost.
>
> We also thank the reviewer for the feedback on clarity and the formatting issue, we will incorporate these in the next version.

---

> > ### Comment · Reviewer_Mhub · 2025-04-04
> >
> > Thanks for the clarification and additional results.
> >
> > Before I give some further comments, could the authors double-check the columns of OBQA and HS of the last table? The numbers do not look right to me. I guess the results of these 2 columns are somehow mixed up.

---

> > > ### Author Response · Authors · 2025-04-04
> > >
> > > Yes I mixed them up when I copied the number from my spreadsheet to here. In the second column of the last table, Hellaswag (HS) should be 79.05 and OpenBookQA (OBQA) should be 45.00 (the average stays the same). I double-checked other numbers and they are correct. Thanks for pointing that out!
> > >
> > > We also address one remaining question from the initial review phase that we didn't get to during the initial rebuttal:
> > >
> > > > Is it possible to replace the transformer block with ladder transformer in ViT? If so, is there any corresponding experiments?
> > >
> > > We believe so, and in fact, ladder-residual can be applied to almost any popular architecture (eg: ViT, mamba, mixture-of-experts) to overlap communication and computation due to the inherent sequential computation nature in all these architectures. In this paper, we only focus on the language domain and experiment with the transformer-based language model, we are looking forward to extending ladder-residual to more architectures and domains in the future.

---

### Decision · Program_Chairs · 2025-05-01

**Decision:**

Accept (poster)

**Comment:**

This paper proposes an architecture modification that is claimed to allow overlapping of communication and computation. The main idea is to decouple the computation of a layer and the residual sum, which enables overlapping one communication step with the next computation stage. The architecture is defined by introducing  $y_i = x_{i-1}$ and $X_i=(x_i,y_i)$, then the activations at depth $i+1$ are given by

\\[
X_{i+1} = H_i(X_i) \quad \text{where } \quad H_i(x_i, y_i) = (x_i + h(y_i), x_i).
\\]

From this formulation, it becomes more obvious how successive computations can be overlapped. I think the connection with TP/AAR (all-reduce) could be slightly toned down — this is more relevant for the ML systems community and may not be the primary focus of ICML.

That said, the paper demonstrates that the method performs well compared to standard Transformers. It smoothly utilizes hardware acceleration both at inference and training time and can be easily incorporated into a standard Transformer. While the method is simple and doesn’t rely on any significant implementation or architectural tricks, it still achieves strong results.

The reviewers find this method to be of interest. Nevertheless, Reviewer Wjzo noted that the authors could have done a better job citing and comparing their method with existing work, and I agree with this assessment. Relevant prior work should be properly incorporated and discussed, a point that was only lightly addressed in the rebuttal. The reviewer also noted a small discrepancy in results, where the proposed method sometimes underperforms, which could be better discussed.

Regarding this last point: given the simplicity of the trick, the fact that no specific hyper-parameter adaptation were used, and the likelihood that current training methods are stuck in local minima of hyper-parameter configurations, I don’t blame the authors. This game of tuning is the one everybody is playing and is probably improductive for research.

Overall, I believe this is a worthwhile contribution to ICML.